# Depressive symptoms and workplace-violence-related risk factors among otorhinolaryngology nurses and physicians in Northern China: a cross-sectional study

Huiying Fang,[1] Xiaowen Zhao,[2] Haicheng Yang,[3] Peihang Sun,[1] Ying Li,[1] Kexin Jiang,[1] Peng Li,[1] Mingli Jiao,[1,4] Ming Liu,[5] Hong Qiao,[6] Qunhong Wu[7]

HF, XZ and HY contributed equally.

For numbered affiliations see end of article.

**Correspondence to**
Dr Mingli Jiao;
minglijiao@126.com,
Prof. Ming Liu;
liuming6293@126.com,
Ms. Hong Qiao;
qiaoh0823@sina.com and
Dr. Qunhong Wu;
qunhongwu@126.com

## ABSTRACT

**Objectives** Workplace violence is relatively frequent among medical professionals who work in otorhinolaryngology units. This phenomenon reduces the quality of provided medical care and increases the incidence of depressive symptoms among physicians and nurses, seriously affecting their job satisfaction and work efficiency with a negative attitude towards providing treatment. Few existing studies have assessed workplace-violence-related factors associated with depressive symptoms among otorhinolaryngology physicians and nurses.

**Methods** We conducted a cross-sectional study in grade A tertiary hospitals of Heilongjiang province in Northern China, to evaluate the occurrence and level of depressive symptoms among otorhinolaryngology physicians and nurses and to analyse the relationship between them and workplace-violence-related risk factors and demographic variables.

**Results** Of all our participating professionals, (379 otorhinolaryngologists and 273 nurses), 57.2% were found to have depressive symptoms, whereas, of the respondents who had suffered from physical violence, 71.25% had depressive symptoms. Professionals with less than 1 year of experience, as well as professionals who more frequently worked alone, were more likely to suffer from depressive symptoms than their colleagues.

**Conclusions** This research addresses an emerging issue of clinical practice, and its results differ from those of previous studies; specifically, it indicates that the frequency of depressive symptoms among otorhinolaryngology physicians and nurses may be influenced by physical violence, the number of coworkers they have for more than half of their working hours and other workplace-violence-related factors. To reduce the depressive symptoms caused by workplace violence and improve the quality of medical services, medical institutions should implement effective measures to prevent the occurrence of physical violence, strengthen team cooperation ability and increase peer support.

### Strengths and limitations of this study

► The sample size, which is based on the number of grade A tertiary hospitals in Heilongjiang province, may limit the generalisability of this study's findings.

► The quality of the participants' self-recall of stigma and shame of the events of the past 12 months may result in an underestimation of the number of violent incidents.

► Despite the limitations, this study can draw hospital managers' attention to the workplace violence that occurs in otorhinolaryngology departments, motivating them to reduce the occurrence of depressive symptoms in otorhinolaryngologists and endeavour to improve the quality of their medical service.

## INTRODUCTION

In October 2013,[1] a patient at the Zhejiang Provincial People's Hospital of Otorhinolaryngology in China stabbed to death the physician who was providing diagnosis and treatment to him. Similarly, in February 2014, an otorhinolaryngologist at a hospital in Qiqihar, Heilongjiang province, China, was beaten to death while examining a patient. The frequency and severity of such workplace violence (WPV) towards medical professionals in China has increased in recent years, as has its public attention.[2] WPV refers to the physical and psychological harm that professionals may face while performing their duties.[3 4] WPV is a significant increasing phenomenon in healthcare settings, and all health workers, especially nurses, are at risk of suffering aggressive assaults.[5] In fact, the number of WPV incidents reported in healthcare facilities accounts for almost one-quarter of the total number of violent incidents reported for all workplaces,[6]



making healthcare workers the most vulnerable group to WPV.[7] Naturally, the occurrence of hospital violence may affect the health of physicians and nurses and hinder their professional performance, resulting in a negative impact on patient services and overall health.[8–10]

Instances of such violence can be found in several hospital departments,[11] and many studies have focused on the high incidence of violence in emergency rooms.[12 13] However, we found a comparable rate of violent incidences in otorhinolaryngology departments. The China Hospital Association 2014 data show that from 2008 to 2012, incidences of severe hospital violence increased from 47.7% to 63.7% of all hospital violence[14] and, on examining 30 cases of physicians and nurses who were victims of fatal WPV in China over the past 10 years, we determined that professionals working in otorhinolaryngology departments account for over 10% of the total.

The main causes of violence towards otorhinolaryngology physicians and nurses include the following: (1) patients tend to consider otorhinolaryngology-related diseases as less important, (2) physicians and nurses in this department have limited professional skills compared with those in other departments and (3) treatment results of otorhinolaryngological diseases are not ideal (ie, many conditions cannot be immediately or completely cured).[15] Thus, considering these dangers, it is clear that the working environment of otorhinolaryngology physicians and nurses should be a concern for hospital administrations, both from a mental health and WPV perspective.

Previous studies have shown that WPV is a contributing factor to employees suffering from mental illnesses such as depression.[16–18] It is well known that depression is a common illness, and it seems to be more prevalent in high-income countries; the incidence rate of depression among adults in low-income to middle-income countries is 11.1%, compared with 14.6% in high-income countries[19 20]; specifically, the incidences of depression in the USA, Canada and the UK are 12%, 19.3% and 15.5%, respectively.[21 22] In Canada, one in 10 nurses have been found to have depressive symptoms,[23] and in France the situation is even more serious, with one-third of nurse managers suffering from such symptoms.[24] Depressive symptoms can negatively affect job satisfaction, work efficiency and overall happiness, resulting in a negative attitude towards providing medical treatment. Over the long term, this can affect the development of human resources in health fields and reduce the quality of medical care provided.[9 10 25] Research shows that Chinese physicians and nurses experience various psychological issues, with the incidence of depression determined to be 34.2%.[26] In a study based in Southern China, (specifically, Guangdong province), depressive symptoms were found to be associated with WPV, long working hours, shift frequency and departmental operations.[8] Also notable is the fact that studies have revealed a difference in the proportion of depressive symptoms among nurses in different hospital departments, which strongly suggests that managers should seek to determine the reason for these differences

between departments through the use of their employee relations and other suitable processes.[8] Further, it should also be noted that much of the existing research primarily assesses the overall situations of hospitals, with few studies focusing on specialised departments. Of the latter studies, the focus has primarily been placed on the emergency department; however, as mentioned above, the department of otolaryngology witnesses a similar number of violent phenomena as emergency rooms, and more than that found in other departments; therefore, the occurrence of depressive symptoms among otolaryngology staff should be taken seriously.

Otolaryngology is a key department in most hospitals and has a number of unique characteristics when compared with other departments. First, there are many types of otolaryngology-related diseases, each presenting in unique anatomical locations, and these can affect patients' breathing, speech, smell and hearing. Many such diseases have long cure times; therefore, providing patients with fast and effective treatment is not always possible. Second, the patient population of an otolaryngology department can span a wide range of age groups, resulting in relatively large workloads for physicians and nurses. Third, otolaryngology involves a variety of specialisations, requiring physicians and nurses to engage in continual learning to improve their professional and technological skills.[15 27]

For these reasons, we hypothesise that the proportion of otolaryngology physicians and nurses suffering from depressive symptoms exceeds that of other hospital departments. While some scholars have studied the influencing factors depressive symptoms have on physicians and nurses in general, few existing studies have assessed WPV-related factors associated with depressive symptoms among otorhinolaryngology physicians and nurses. Consequently, the purpose of this study is: (1) to investigate the prevalence of depressive symptoms among otorhinolaryngology physicians and nurses and (2) to assess the risk factors that influence depressive symptoms, including WPV-related risk factors and demographic variables, and to present some specific suggestions.

## METHODS
### Study population
We conducted a retrospective cross-sectional survey of otorhinolaryngologists and nurses from grade A tertiary hospitals in Heilongjiang province. Grade A tertiary hospitals are classified in accordance with China's current 'hospital classification management approach', which is designed to implement the division of medical institutions. When a medical institution has over 501 beds, it has the capability to provide high-level, specialised medical and health services in several areas, and also to implement higher-level education and scientific research tasks; such hospitals are ranked as 'tertiary'. Further, according to the scoring standard, those that score more than 900 points as general hospitals are categorised into

the highest classification level for hospitals in mainland China.[28]

## Data collection

Considering the length of time allotted for this research and the limited time available to clinical staff, we decided to adopt a web-based questionnaire approach, which would help to ensure the integrity and effectiveness of the data collection, and we also chose to apply a convenience sampling method. First, with the help of the chairman of the Heilongjiang province branch of the Association of Otorhinolaryngology Physicians, we obtained approval from the Heilongjiang Provincial Association of Otorhinolaryngology to conduct the study. The association consequently provided a list of grade A tertiary hospitals and associated physicians' email addresses, which allowed us to contact otolaryngology directors to seek assistance in acquiring related nurses' email addresses. We then sent these individuals an introductory email with a link to the informed consent form and questionnaire. After the initial email was sent, a reminder email was sent to any physicians and nurses who did not respond initially. Through this process, we obtained 652 valid questionnaires, (total efficiency of 83.6%). The data were collected from 10 January 2017, to 15 February 2017.

## Questionnaire preparation

After obtaining written permission, we sent the respondents a questionnaire that was developed by the International Labour Office, International Council of Nurses, WHO and Public Service International in 2003 with the goal of measuring violence in the workplace.[29] We translated the questionnaire into Mandarin and then back into English to verify the accuracy of the Mandarin version. We then invited the aforementioned experts identified by the otorhinolaryngology association to modify the questions as needed, and then assessed the validity of the contents of the questionnaire, including its suitability for Chinese culture and appropriateness of the translation. These experts included epidemiologists, health-management specialists, public health specialists, health system specialists, otolaryngology experts and administrative experts (15 persons). We also conducted a 2-week retest reliability check (Cronbach's α=0.87) by having a group of 42 otorhinolaryngology physicians and nurses from five grade-A tertiary hospitals take the test, and then take it again 2 weeks later.

The final version of the questionnaire included four parts: (1) the demographic characteristics of the professionals (gender, age, marital status, years of work experience, education level, occupation, seniority) and workplace-related conditions, number of patients received per day, shift work, number of coworkers for more than half of working hours and anxiety experienced at work (which was rated from 1 (absent) to 5 (extremely high)); (2) experience of physical violence in the previous 12 months (defined as intentional injury to body); (3) experience of non-physical violence over the previous 12 months (including verbal abuse, threatening events and sexual harassment) and (4) depressive symptoms, measured using a Chinese version of the Zung Self-Rating Depression Scale (SDS), which has good reliability. SDS is usually used to measure the level of depressive symptoms in the general population over the previous week and is widely used in research into WPV and depressive symptoms.[8 30] Specifically, it facilitates the revealing of subjective feelings using 20 items which are rated in terms of severity using a four-point scale ranging from 1 (absent or minimum) to 4 (high or extremely high). The total score was recorded as the original score, the scores for the 20 items were added together and finally the resulting score was multiplied by 1.25 to get the standard score. A standard score of 53–62 points represents mild depression, 63–72 points represents moderate depression and >72 points signifies severe depression. To ensure that the results of this study can be compared with the results of related studies, we chose to use the same classification method as other studies: setting a final score of ≥53 points as indicating the presence of depressive symptoms.[31 32]

## Data analysis

Descriptive statistical calculations were conducted on the respondents' demographic characteristics and their reported frequency of experiencing depressive symptoms; in addition, a χ2 test was used to analyse the correlation between symptoms of depression and the respondents' characteristics, as well as WPV-related factors. Further, logistic regression analysis was used to determine the characteristics and factors (including years of work experience, education level, seniority, number of patients received per day, number of coworkers for more than half of working hours, anxiety experienced at work and physical violence) which were likely to predict depression. SPSS V.19.0 2010 (IBM) was used to conduct the analysis. Statistical significance is indicated by P<0.05.

## Ethics statement

All respondents received an informed consent form, which described the goal and method of the data collection, how the data would be handled and also an assurance of confidentiality. All respondents who gave their informed consent completed the questionnaire.

## RESULTS

In table 1, the demographic characteristics of the study participants are shown. Of these otolaryngology staff, 63.7% were women, almost half (41.7%) were between 30 and 39 years of age, and three-quarters (78.1%) were married. The majority of respondents (69.8%) had at least 10 years of experience in otolaryngology and 78.2% were graduated or specialised. Of the respondents, 19.5% were senior within the department.

Descriptive statistics concerning the depressive symptoms of the participants are presented in table 2; notably, the table shows that 57.2% of the 652 otorhinolaryngology

**Table 1** Participant demographic characteristics (n=652)

| Characteristic | n (%) |
|---|---|
| Gender | |
| Male | 237 (36.3) |
| Female | 415 (63.7) |
| Age | |
| ≤29 | 214 (32.8) |
| 30–39 | 272 (41.7) |
| ≥40 | 166 (25.5) |
| Marital status | |
| Unmarried | 130 (19.9) |
| Married | 509 (78.1) |
| Divorced | 9 (1.4) |
| Widowed | 1 (0.2) |
| Other | 3 (0.5) |
| Years of work experience | |
| <1 | 76 (11.7) |
| 1–10 | 379 (58.1) |
| 11–20 | 126 (19.3) |
| 21–30 | 61 (9.4) |
| >30 | 10 (1.5) |
| Seniority | |
| Senior | 127 (19.5) |
| Intermediate | 237 (36.3) |
| Primary | 168 (25.8) |
| No title (formal doctor) | 65 (10.0) |
| Interns (resident training in hospital or postgraduate) | 55 (8.4) |
| Education level | |
| Graduate and above | 130 (19.9) |
| Undergraduate | 380 (58.3) |
| Community college | 97 (14.9) |
| Vocational school | 40 (6.1) |
| Other | 5 (0.8) |
| Occupation | |
| Physician | 379 (58.1) |
| Nurse | 273 (41.9) |

physicians and nurses had depressive symptoms. Moreover, approximately half (49.54%) of the respondents worked with one or two colleagues for more than half of their working hours while, of the respondents with depression symptoms, three-quarters (76.41%) had experienced varying degrees of anxiety. Additionally, of the respondents who had suffered physical violence, 71.25% had depressive symptoms. We then assessed associations between sample characteristics and depressive symptoms using Pearson $\chi^2$ tests and determined that years of work experience, seniority, number of coworkers for more than half of the working hours, anxiety experienced at work and physical violence are significantly associated with depressive symptoms.

In table 3, the variables related to depressive symptoms, determined using a multiple logistic model, an adjusted OR and a 95% CI, are shown. Our otorhinolaryngology physicians and nurses who suffered from physical violence (OR=1.82, 95% CI 1.06 to 3.12) were more likely to suffer from depressive symptoms than those who had not. Moreover, participants with less than 1 year of experience were also more likely to suffer from depressive symptoms, while those with more experience were progressively less likely to suffer from depressive symptoms. Otorhinolaryngology physicians and nurses who worked alone for more than half of their working hours were more likely to suffer from depressive symptoms than those who worked frequently with other colleagues. Relative to the seniority of otorhinolaryngology physicians and nurses, formal staff and those with junior and intermediate titles showed protective factors for depressive symptoms.

## DISCUSSION

Our study examined the association between depressive symptoms and WPV-related risk factors with regard to otorhinolaryngology physicians and nurses in Northern China. Our results confirm that depressive symptoms are common among professionals in this department, with the incidence of depressive symptoms shown to be 57.2%. There is limited research on depression among otorhinolaryngology staff, so we could not directly compare our results with those of other studies; however, these results show a significantly higher prevalence of depressive symptoms than that found among general physicians and nurses in the USA (11.3%[33] and 41%, respectively),[34] general physicians in the Netherlands (29%),[35] and Iranian physicians and nurses (28.76%).[36] Further, this number is also slightly higher than the incidence of depressive symptoms reported among emergency department nurses in Taiwan.[1] In addition, the studies of Yang and Hu, and Ji et al[37 38] reported that the incidence of depressive symptoms among healthcare workers in China across all hospital departments is 45.3% and 47.2%, respectively; however, the incidence of depression differs between departments. Otorhinolaryngology physicians and nurses have a higher incidence of depressive symptoms than most other departments outside of emergency, and also report higher than normal levels[8 30]; however, it should be noted that few previous studies have focused on the incidence of depressive symptoms among these professionals in China. Unfortunately, in China, the frequency of violence (which is a major predictor of depression) within otorhinolaryngology department is relatively high due to various factors involving treatment options and professional skill levels[15]; therefore, research into the association between depressive symptoms and WPV-related risk factors is essential for preserving the mental health of practitioners and ensuring quality of care.

**Table 2** Descriptive statistics associated with depressive symptoms of the participants

| Variables | Non-depressive tendency n=279 (42.8%) n (%) | Depressive tendency n=373 (57.2%) n (%) | $\chi^2$ | P |
|---|---|---|---|---|
| Gender | | | 0.01 | 0.92 |
| Male | 102 (36.56) | 135 (36.19) | | |
| Female | 177 (63.44) | 238 (63.81) | | |
| Age | | | 0.20 | 0.91 |
| ≤29 | 94 (33.69) | 120 (32.17) | | |
| 30–39 | 114 (40.86) | 158 (42.36) | | |
| ≥40 | 71 (25.45) | 95 (25.47) | | |
| Marital status | | | 4.92 | 0.30 |
| Unmarried | 49 (17.56) | 81 (21.72) | | |
| Married | 226 (81.00) | 283 (75.88) | | |
| Divorced | 2 (0.72) | 7 (1.88) | | |
| Widowed | 0 (0) | 1 (0.27) | | |
| Other | 2 (0.72) | 1 (0.27) | | |
| Years of work experience | | | 10.26 | 0.04 |
| <1 | 33 (11.83) | 43 (11.53) | | |
| 1–10 | 163 (58.42) | 216 (57.91) | | |
| 11–20 | 48 (17.20) | 78 (20.91) | | |
| 21–30 | 26 (9.32) | 35 (9.38) | | |
| >30 | 9 (3.23) | 1 (0.27) | | |
| Seniority | | | 14.00 | 0.01 |
| Senior | 52 (18.64) | 75 (20.11) | | |
| Intermediate | 89 (31.90) | 148 (39.68) | | |
| Primary | 69 (24.73) | 99 (26.54) | | |
| No title (formal doctor) | 39 (13.98) | 26 (6.97) | | |
| Interns (resident training in hospital or postgraduate) | 30 (10.75) | 25 (6.70) | | |
| Education level | | | 7.32 | 0.12 |
| Graduate and above | 52 (18.64) | 78 (20.91) | | |
| Undergraduate | 174 (62.37) | 206 (55.23) | | |
| Community college | 31 (11.11) | 66 (17.69) | | |
| Vocational school | 20 (7.17) | 20 (5.36) | | |
| Others | 2 (0.72) | 3 (0.80) | | |
| Occupation | | | 0.45 | 0.50 |
| Physician | 158 (56.63) | 221 (59.25) | | |
| Nurse | 121 (43.37) | 152 (40.75) | | |
| Number of patients received per day | | | 1.02 | 0.91 |
| 1–20 | 122 (43.73) | 154 (41.29) | | |
| 21–30 | 83 (29.75) | 116 (31.10) | | |
| 31–40 | 39 (13.98) | 55 (14.75) | | |
| 41–50 | 13 (4.66) | 22 (5.90) | | |
| >50 | 22 (7.89) | 26 (6.97) | | |
| Shift work | | | 0.01 | 0.92 |
| Yes | 222 (79.57) | 298 (79.89) | | |
| No | 57 (20.43) | 57 (15.28) | | |

Continued

**Table 2** Continued

| Variables | Non-depressive tendency n=279 (42.8%) n (%) | Depressive tendency n=373 (57.2%) n (%) | $\chi^2$ | P |
|---|---|---|---|---|
| Number of coworkers for more than half of the working hours | | | 13.62 | 0.01 |
| 1 | 17 (6.09) | 40 (10.72) | | |
| 2 | 107 (38.35) | 159 (42.63) | | |
| 3–5 | 97 (34.77) | 129 (34.58) | | |
| 6–10 | 35 (12.54) | 32 (8.58) | | |
| >10 | 23 (8.24) | 13 (3.49) | | |
| Anxiety experienced at work | | | 9.54 | 0.04 |
| Absent | 89 (31.90) | 88 (23.59) | | |
| Low | 52 (18.64) | 61 (16.35) | | |
| Moderate | 74 (26.52) | 105 (28.15) | | |
| High | 23 (8.24) | 38 (10.19) | | |
| Extremely high | 41 (14.70) | 81 (21.72) | | |
| Physical violence | | | 3.34 | 0.01 |
| Yes | 23 (8.24) | 57 (15.28) | | |
| No | 256 (91.76) | 316 (84.72) | | |
| Non-physical violence | | | 3.43 | 0.06 |
| Yes | 91 (32.62) | 148 (39.68) | | |
| No | 188 (67.38) | 225 (60.32) | | |

This study found several factors associated with the occurrence of depressive symptoms. First, years of work experience and seniority were identified; the fewer a physician's or nurse's years of work experience, the more likely they are to suffer from depressive symptoms. This has also been confirmed in previous studies on healthcare professionals in China.[39] Less experienced physicians and nurses often receive insufficient social support and are more likely to be affected by occupational stress.[39 40] Thus, the mental state of physicians and nurses with few years of experience should be evaluated as early as possible in order to provide them with preventative and proactive psychological support, as well as training opportunities.[41]

Second, considering WPV-related risk factors, number of coworkers for more than half of working hours was found to be a predictor of depression among otorhinolaryngology professionals. The results of the survey showed that, when compared with physicians and nurses who mainly worked with two colleagues, those who frequently worked alone were more likely to suffer from depressive symptoms; the higher the number of colleagues working together, the lower the risk of depressive symptoms. Working alongside colleagues can help physicians and nurses handle pressure, dissatisfaction and other emotional fluctuations. Previous studies have also shown that social support can reduce the incidence of depressive symptoms, help individuals manage stress and promote good mental health.[42] Peer support intervention, in particular, can effectively reduce depressive symptoms, not only because it can be implemented

through daily interactions, but also because it can have the same effect as group cognitive behavioural therapy.[43] Support at work is a social-network resource associated with the work environment, and employees at all levels can benefit from the support of their colleagues. In the demand–control–support model, it has been highlighted that social support is similar to job control, and may have a buffering or reinforcing effect on individuals' psychological responses; for example, work friendship between colleagues has been reported to be an important social support.[44] Therefore, the working status of medical staff (eg, time spent working alone) should be considered in order to strengthen peer support (as well as team cooperation), and thereby effectively prevent and improve depressive symptoms.

Third, this study revealed that physical violence is an important predictor of incidences of depressive symptoms within the otolaryngology department. Our findings suggest that the risk of physicians and nurses who have suffered physical violence developing depressive symptoms is 1.82 times greater than that for professionals who have not experienced such violence. This is supported by previous surveys, which have also indicated that WPV is a strong predictor of psychological problems such as depressive symptoms.[16] Depression is one of several negative yet common consequences experienced by physicians and nurses who suffer from WPV.[45] Among them, physical violence is more likely to cause physicians and nurses to experience depressive symptoms.[39] The reasons for the high rate of physical violence within otolaryngology

**Table 3** Multiple logistic regression of depressive and non-depressive symptoms

| | OR | 95% CI | P |
|---|---|---|---|
| Years of work experience | | | |
| <1 | 1.00 | Reference | |
| 1–10 | 0.04 | 0.01 to 0.39 | 0.01 |
| 11–20 | 0.06 | 0.01 to 0.52 | 0.01 |
| 21–30 | 0.06 | 0.01 to 0.52 | 0.01 |
| >30 | 0.07 | 0.01 to 0.66 | 0.02 |
| Education level | | | |
| Graduate and above | 1.00 | Reference | |
| Undergraduate | 1.32 | 0.20 to 8.89 | 0.78 |
| Community college | 1.38 | 0.21 to 8.85 | 0.74 |
| Vocational school | 0.67 | 0.10 to 4.46 | 0.68 |
| Others | 1.45 | 0.21 to 10.13 | 0.71 |
| Seniority | | | |
| Senior | 1.00 | Reference | |
| Intermediate | 0.37 | 0.16 to 0.85 | 0.02 |
| Primary | 0.35 | 0.17 to 0.73 | 0.01 |
| No title (formal doctor) | 0.48 | 0.23 to 0.99 | 0.04 |
| Interns (resident training in hospital or postgraduate) | 0.92 | 0.40 to 2.15 | 0.85 |
| Number of patients received per day | | | |
| 1–20 | 1.00 | Reference | |
| 21–30 | 1.03 | 0.52 to 2.06 | 0.93 |
| 31–40 | 0.98 | 0.48 to 1.98 | 0.94 |
| 41–50 | 0.88 | 0.40 to 1.93 | 0.75 |
| >50 | 0.71 | 0.27 to 1.87 | 0.49 |
| Number of coworkers for more than half of the working hours | | | |
| 1 | 1.00 | Reference | |
| 2 | 0.27 | 0.11 to 0.70 | 0.01 |
| 3–5 | 0.41 | 0.20 to 0.89 | 0.02 |
| 6–10 | 0.41 | 0.19 to 0.91 | 0.03 |
| >10 | 0.73 | 0.30 to 1.74 | 0.47 |
| Anxiety experienced at work | | | |
| Absent | 1.00 | Reference | |
| Low | 1.84 | 1.09 to 3.09 | 0.02 |
| Moderate | 1.61 | 0.91 to 2.83 | 0.10 |
| High | 1.30 | 0.78 to 2.15 | 0.31 |
| Extremely high | 1.13 | 0.58 to 2.21 | 0.72 |
| Physical violence | | | |
| Yes | 1.82 | 1.06 to 3.12 | 0.03 |
| No | 1.00 | Reference | |

departments are varied, but an important factor is that otorhinolaryngology-related diseases are often complex and involve longer courses of treatment, which can lead to patient insomnia, anxiety and other psychological problems, which can in turn lead to a loss of trust in physicians and nurses. As such, these patients can be prone to adverse emotions.[46] Further, due to the nature of many otorhinolaryngology diseases, which affect speech and hearing, patients are often unable to effectively express their dissatisfaction, making them more inclined to choose physical violence to express their discomfort or psychological distress.[47]

Physical violence can also cause psychological damage to physicians and nurses; for example, it can increase levels of anxiety, which can result in poor productivity, lower quality of work, a decline in individual sense of accomplishment and poor relationships with patients, managers and colleagues.[12 47 48] In addition, physical violence can also have a negative impact on physicians and nurses' social relationships both inside and outside the hospital.[48] Consequently, the relationship between physicians and nurses and their patients should be improved through different measures, including by attaching importance to patients' psychological status during clinical diagnosis and treatment processes, enhancing the communication skills of otorhinolaryngology staff and improving patients' awareness of their disease and potential treatment options (including duration and adverse side-effects). Concurrently, medical institutions should adopt a 'zero tolerance' approach to violent incidents and, to reduce the harm caused by physical violence, managers should strengthen the provision of timely psychological comfort and psychological counselling for employees who have experienced physical violence.

In previous studies, the incidence of depressive symptoms was associated with physicians and nurses' workload and shift characteristics[49–53]; however, this study did not find any relevant association between depression and these two factors among otorhinolaryngology professionals, similar to the findings of another study of resident physicians in Turkey.[54] Future research should further explore this area.

This study has the following limitations. First, we only investigated otorhinolaryngology physicians and nurses working in grade A tertiary hospitals in Heilongjiang province, so the results may not be fully representative of the entire situation in Northern China, and cannot be extended to other types of hospitals or departments. Future studies should conduct in-depth exploration of the different types of otorhinolaryngological services at hospitals of different grades. Second, physicians and nurses who were absent from work because of depression were not assessed; thus, the reported rate of depressive symptoms may be lower than the true value. Third, because the respondents were asked to review the events of the past 12 months, the prevalence of violence could have been under-reported as a result of reporting or recall bias or stigma and shame. Fourth, this study focuses on the more distinctive WPV-related factors for this analysis; however, there may be additional factors related to the occurrence of depressive symptoms that could be further explored.

Nevertheless, the results of this study indicate that physical violence and mainly working alone are risk factors for depressive symptoms among otorhinolaryngology physicians and nurses, and this provides some basis and direction for future research.

In conclusion, the results show that otorhinolaryngology physicians and nurses in grade A tertiary hospitals in Heilongjiang province, China, have a high risk of experiencing depressive symptoms. Of the 652 respondents, 57.2% reported varying degrees of depressive symptoms. Our findings also show that experiencing physical violence, number of coworkers for more than half of working hours and other WPV-related factors are associated with symptoms of depression. Therefore, on one hand, it is clear that medical institutions should pay more attention to otorhinolaryngology. Essential measures include improving communication between medical staff and patients and the adoption of a 'zero tolerance' attitude towards hospital violence. On the other hand, attention should be paid to the construction of team relationships, improving teams' cooperation and giving full backing to peer support to alleviate depressive symptoms. Finally, more attention should also be paid to lateral violence (bullying, harassment, physical violence between staff, etc) to address depressive symptoms caused by such violence. However, as we did not study the impact of lateral violence on depressive symptoms, this represents an avenue of research for future studies.

**Author affiliations**
[1]Department of Health, Policy and Hospital Management, School of Public Health, Harbin Medical University, Harbin, China
[2]Department of Health Economy, School of Public Health, Harbin Medical University, Harbin, China
[3]Department of Neurosurgery, The 2nd Affiliated Hospital of Harbin Medical University, Harbin, China
[4]Institute of Quantitative and Technical Economics, Chinese Academy of Social Science, Beijing, China
[5]Department of Otorhinolaryngology, The 2nd Affiliated Hospital of Harbin Medical University, Harbin, China
[6]Endocrine and Metabolic Diseases, The 2nd Affiliated Hospital of Harbin Medical University, Harbin, China
[7]Department of Social Medicine, School of Public Health, Harbin Medical University, Harbin, China

**Contributors** HF, XZ and HY drafted the manuscript. MJ designed the study. PS, YL, KJ and PL collected the data. MJ, ML, HQ and QW analysed the data. MJ contributed to the revision of the manuscript. All authors approved the final manuscript for publication.

**Funding** This study was funded by the Natural Science Foundation of China (Grant No. 71273002, 71473064,71673073); New Century Excellent Talents of University from the Ministry of Education, China (Grant No.1252-NCET02); the China Postdoctoral Science Foundation (2015M570211, 2016T90181); the Heilongjiang Provincial Association of Social Sciences (15058), and the Collaborative Innovation Centre of Social Risks Governance in Health.

**Competing interests** None declared.

**Patient consent** Obtained.

**Ethics approval** The study protocol was reviewed and approved by the Research Ethics Committee of Harbin Medical University.

**Provenance and peer review** Not commissioned; externally peer reviewed.

**Data sharing statement** Data are available from the corresponding author on request.

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
