## [Reviewer comments · BMJ Open]

ARTICLE DETAILS

TITLE (PROVISIONAL)	Depressive symptoms and workplace violence-related risk factors among otorhinolaryngology nurses and physicians in northern China: A cross-sectional study
AUTHORS	Fang, Huiying; Zhao, Xiao; Yang, Hai; Sun, Peihang; Li, Ying; Jiang, Ke; peng, li; Jiao, Mingli; Liu, Ming; Qiao, Hong; Wu, Qunhong

VERSION 1 – REVIEW

REVIEWER	Ferri Paola University of Modena and Reggio Emilia, Italy
REVIEW RETURNED	10-Oct-2017

GENERAL COMMENTS	This manuscript deals with an emerging issue of clinical practice, the relationship between workplace violence and depression of professionals and the quality care provided. This manuscript deals with an emerging issue of clinical practice, the relationship between workplace violence and depression of professionals and the quality care provided. In my opinion, the topic is relevant and has been managed in a scientific way. Nevertheless, the manuscript needs minor revisions, concerning in particular the language, that has to be revised according to the suggestions of an English mother-tongue speaker. Moreover, the terms should be more appropriate and many observations reported in more details, according to a scientific language. The Abstract needs substantial revisions in language and contents. I suggest the following modifications: Objectives: Workplace physical violence is relatively frequent among medical professionals who work in otorhinolaryngology units. This phenomenon not only reduces the quality of provided medical care, but also increases the incidence of depressive symptoms among physicians and nurses, seriously affecting their job satisfaction and work efficiency with a negative attitude toward providing treatment. Few existing studies analysed otorhinolaryngology physicians and nurses suffering from depression and related factors. Methods: We conducted a cross-sectional study in tertiary (it is not clear the meaning of tertiary) hospitals of Heilongjiang Province in northern China, to evaluate the occurrence and level of depressive symptoms among professionals and to analyse the relationship between them and workplace violence-related risk factors or other selected professional variables.
--

Results: 57.2% of all our participating professionals (379 otorhinolaryngologists and 273 nurses) was found to have depressive symptoms, whereas, of the respondents who had suffered from physical workplace violence, 71.25% had depressive symptoms. Professionals with less than 1 year of experience as well as professionals who more frequently worked alone were more likely to suffer from depressive symptoms than other colleagues.

Conclusions: This research indicates that the mental health of otorhinolaryngology physicians and nurses can be vulnerable to depression especially if they had been exposed to workplace violence. Health care settings should be carefully monitored to prevent the occurrence of physical violence in order to reduce depressive symptoms of health workers and improve quality of care.

In the Introduction, I suggest to add at page 5, line 24 (after "...face while performing their duties [3,4].") the following sentence with this citation (to report in the References):

Workplace violence is a significant increasing phenomenon in health care settings and all healthworkers, especially nurses, are at risk of suffering aggressive assaults (Ferri P, Silvestri M, Artoni C, Di Lorenzo R. Workplace violence in different settings andamong various health professionals in an Italian general hospital: a cross-sectional study. Psychology Research and Behavior Management 2016;9;263–275).

In the Methods, the paragraph "2.1. Study Population and Data Collection", at page 9, lines 9-51, should be shorter. Moreover, an explanation of "Grade A tertiary " should be reported. At page 9, the subheading 2.2 could be changed to "The questionnaire preparation".

At page 10, line 8, the citation should be changed because the article cited is a qualitative research and not included the ILO questionnaire administered in this study. At the following link you can find the ILO questionnaire (I advise to change the reference n. 27 with this link). http://www.who.int/violence_injury_prevention/violence/interpersonal/en/WVquestionnaire.pdf

At page 10, lines 36-54, the sentence should be changed as the following (reporting the same terms in the tables):

"The final version of the questionnaire included four parts: (1) demographic characteristics of professionals (gender, age, marital status, years of work experience, education level, occupation, seniority) and workplace-related conditions [number of patients received per day, shift work, and anxiety experienced at work from 1 (absent) to 5 (extremely high) level]; (2) physical violence in the previous 12 months (defined as intentional injury to body); (3) situation of nonphysical violence in the previous 12months (including verbal abuse, threatening events, and sexual harassment); and (4)....."

At page 11, lines 3-19, I suggest the following modifications: "Depression Scale (SDS), which was able to reveal subjective feelings using 20 items with a severity score ranging from 1 (absent or minimum) to 4 (high or extremely high). The total score was recorded as the original score, then the scores of the 20 items were added, and finally the resulting score was multiplied by 1.25 to get the standard score. A final score of ≥ 53 points indicated the presence of depressive symptoms."

At page 11, line 57, I suggest to modify as follows: All respondents who gave their informed consent filled....

In the Results section, I advise the following changes:

At page 13, line 7, "In Table 1, the demographic characteristics of the study participants are shown."

At page 13, lines 14-19, "...and 78.2% were graduated or specialized. Of the respondents, 19.5% was senior within the department."

At page 13, line 24, "...shown in Table 2. 57.2% of the 652 otorhinolaryngology physicians.."

At page 13, lines 26-29, "...with one or two colleagues for more than half of their working hours."

At page 13, lines 46-49, "In Table 3, the variables related to depressive symptoms at the multiple logistic model, with the adjusted odds ratio (OR) and 95% confidence interval (CI), are shown. Our otorhinolaryngology physicians and nurses who..."

At page 14, lines 5-20, I suggest the following changes:
"Otorhinolaryngology physicians and nurses who worked alone for more than half of their working hours were more likely to suffer from depressive symptoms than those who worked frequently with other colleagues. Relative to the seniority of otorhinolaryngology physicians and nurses, formal staff and those with junior and intermediate titles showed protective factors for depressive symptoms."
It is not clear the meaning of "formal staff".

In the Discussion section, at page 15, lines 54-57 and page 16, lines 3-6, I suggest to cancel the following sentence "In China, the otorhinolaryngology department is a significant and indispensable group within the hospital, as it is related to patients' ear, nose, throat, and even the head and neck." because too generic and not specific. I suggest to modify the following sentence: "Unfortunately, in China, the frequency of violence within otorhinolaryngology department is relatively high due to various factors involving...."

At page 18, line 19 the word "employee" as adjective for anxiety should be cancelled.

At page 18, lines 46-57, I suggest the following changes: "In previous studies, the incidence of depressive symptoms was associated with physicians and nurses' workload and shift characteristics; however, this study did not find any relevant association between depression and these two factors among otorhinolaryngology professionals [46-49], as well as another study of resident physicians in Turkey [50]."
In this regard, I suggest to add the following citation (page 18, line 51, after "shift characteristics"), to report in the References: Ferri P, Guadi M, Marcheselli L, Balduzzi S, Magnani D, Di Lorenzo R. The impact of shift work on the psychological and physical health of nurses in a general hospital: a comparison between rotating night shifts and day shifts. Risk Manag Healthc Policy. 2016 Sep 14;9:203-211.

"Third, this study investigated only the association of workplace violence-related factors with depression without analyzing additional factors related to the occurrence of depressive symptoms."

I suggest to cancel the following sentence because too speculative: "Nevertheless, the current results suggest that otorhinolaryngology physicians and nurses are more susceptible to depressive symptoms than other healthcare professionals and that certain risk factors associated physicians and nurses are more susceptible to depressive with workplace violence have a direct effect on this occurrence."

	At page 19, line 39: I suggest to modify the sentence as following: "In conclusion, .." In Table 2, page 32, line 9, the term "Title" should be changed to "Seniority" In Table 2, page 32, line 19, the term "formal doctor" is not clear. In Table 2, page 33, line 31, "A person" should be changed to "1". In Table 2, page 33, lines 21-29, "Number of colleagues during work" could be changed to "Number of co-workers". In Table 2, page 33, line 46, "Zero" should be changed to "Absent". In Table 3, I suggest to modify the terms as above reported.
--	---

REVIEWER	Andréa Tenório Correia da Silva University of São Paulo, Brazil
REVIEW RETURNED	19-Oct-2017

GENERAL COMMENTS	Comments to the authors This cross-sectional study investigated depressive symptoms among otorhinolaryngology nurses and physicians in tertiary hospitals and their associations with work-related variables, including exposure to workplace violence. It is one of few studies of otorhinolaryngology nurses and physicians in tertiary hospitals. However, the study does not offer any unique insights into job related factors and their relationship with depressive symptoms. The authors should point out what the novelty of the study is, and emphasize how it fills in knowledge gaps. It is essential to explain why the approach is new and important. The objectives of the study should be more accurately described. The way it is displayed throughout the manuscript is inconsistent and may lead the reader to uncertainty about whether the objective was: (1) to investigate the association between exposure to workplace violence and depressive symptoms; (2) to examine variables associated with workplace violence; (3) or to assess factors associated with depressive symptoms, including workplace violence. For instance, in the abstract "to investigate the relationship between workplace violence-related risk factors and depressive symptoms", and in the introduction "we found few previous studies on Chinese otorhinolaryngology physicians and nurses suffering from depression-related factors, including violence. This study seeks to assess the level of depression... and to identify associations among depressive symptoms and workplace violence-related risk factors". With respect to the methods section, the questionnaire used to measure depression can make comparison with other studies limited. Moreover, it is necessary to explain the use of the Zung Self-Rating Depression Scale to assess both depression and anxiety, and classifying participants in levels of anxiety (zero, low, moderate, high, and extremely high). In addition, since this is a cross-sectional study, it is not possible to examine prediction. My suggestion to the authors would be to review this section according to the items in the STROBE Statement. This will probably make it easier for the audience to understand what the authors have done step-by-step. The limitations should include: (1) physicians and nurses who were absent from work because of depression were not assessed, and (2) the prevalence of violence could have been under-reported due to reporting or recall bias, or stigma and shame.
--

	Finally, the conclusion should be more focused on the actual results. The main finding was the association of exposure to physical violence and depressive symptoms. The authors suggested that intervention should include improving team relationships. Nevertheless, the authors have not addressed workplace conflicts (such as bullying, harassment, physical violence between staff etc.), or social support from colleagues and supervisors (e.g. Demand-Control-Support model).
--	--

VERSION 1 – AUTHOR RESPONSE

Replies to Reviewer 1:

Dear Prof. Ferri Paola,

First of all, thank you for your earnest and constructive sincere comments on this article. According to your suggestion, I made a careful revision of the article, and gave a detailed answer to the questions you asked, and at the same time, the language was further polished. I don't know if I understand your proposal correctly. If you have any other questions, please do not hesitate to contact me. Thank you! Here's the details. (The modified part of the manuscript is marked with yellow.)

(1) The Abstract needs substantial revisions in language and contents. I suggest the following modifications:

Answer: In the abstract section, according to your opinion, I have made a careful revision. (Page 3 to 4) On the basis of your suggestion, the conclusion is supplemented, and the innovation and research purpose of this research are emphasized. The revised contents are as follows.

Objectives: Workplace violence is relatively frequent among medical professionals who work in otorhinolaryngology units. This phenomenon not only reduces the quality of provided medical care, but also increases the incidence of depressive symptoms among physicians and nurses, seriously affecting their job satisfaction and work efficiency with a negative attitude toward providing treatment. Few existing studies have assessed workplace-violence-related factors associated with depressive symptoms among otorhinolaryngology physicians and nurses.

Methods: We conducted a cross-sectional study in Grade A tertiary hospitals of Heilongjiang Province in northern China, to evaluate the occurrence and level of depressive symptoms among otorhinolaryngology physicians and nurses and to analyse the relationship between them and workplace-violence-related risk factors and demographic variables.

Results: Of all our participating professionals, (379 otorhinolaryngologists and 273 nurses), 57.2% were found to have depressive symptoms, whereas, of the respondents who had suffered from physical violence, 71.25% had depressive symptoms. Professionals with less than one year of experience, as well as professionals who more frequently worked alone, were more likely to suffer from depressive symptoms than their colleagues.

Conclusions: This research addresses an emerging issue of clinical practice, and its results differ from those of previous studies; specifically, it indicates that the frequency of depressive symptoms among otorhinolaryngology physicians and nurses may be influenced by physical violence, the number of co-workers they have for more than half of their working hours, and other workplace-violence-related factors. To reduce the depressive symptoms caused by workplace violence and improve the quality of medical services, medical institutions should implement effective measures to prevent the occurrence of physical violence, strengthen team cooperation ability, and increase peer support.

(2) In the Introduction, I suggest to add at page 5, line 24 (after "...face while performing their duties [3,4].") the following sentence with this citation (to report in the References):

Workplace violence is a significant increasing phenomenon in health care settings and all healthworkers, especially nurses, are at risk of suffering aggressive assaults(Ferri P, Silvestri M, Artoni C, Di Lorenzo R. Workplace violence in different settings and among various health professionals in an Italian general hospital: a cross-sectional study. Psychology Research and Behavior Management 2016;9:263–275).

Answer: In the introduction section, according to your suggestion, I added the content and reference that you recommended.(Page 6, line9, It has been marked yellow)

.....duties.3, 4 WPV is a significant increasing phenomenon in health care settings, and all health workers, especially nurses, are at risk of suffering aggressive assaults.5

(3) In the Methods, the paragraph "2.1. Study Population and Data Collection", at page 9, lines 9-51, should be shorter. Moreover, an explanation of "Grade A tertiary " should be reported. At page 9, the subheading 2.2 could be changed to "The questionnaire preparation".

Answer: 1.According to your revised opinion, we revised the manuscript (Page 9 and 10, It has been marked yellow) , "2.1. Study Population and Data Collection" were divided into "2.1 Study Population" and "2.2 Data Collection" two parts.

2.At the same time, the introduction of the Grade A tertiary hospitals were introduced into the "2.1 Study Population".(Page 9, line18 It has been marked yellow) Details about this are as follows.

The Grade A tertiary hospitals is in accordance with China's current "hospital classification management approach" provisions for the division of medical institutions, When the number of medical institutions is more than 501 beds, it is possible to provide high-level specialized medical and health services to several areas and to carry out higher education and scientific research tasks, and according to the score of more than 900 points of the general hospital, It is the highest level of classification for hospitals in mainland China.

(4) At page 10, line 8, the citation should be changed because the article cited is a qualitative research and not included the ILO questionnaire administered in this study. At the following link you can find the ILO questionnaire (I advise to change the reference n. 27 with this link).
http://www.who.int/violence_injury_prevention/violence/interpersonal/en/WVquestionnaire.pdf

Answer: We think carefully about your proposal and find it very reasonable. Therefore, according to your suggestion, we have revised the references.(Page 11, line 3, It has been marked yellow)

(5) 5.1 At page 10, lines 36-54, the sentence should be changed as the following (reporting the same terms in the tables):

"The final version of the questionnaire included four parts: (1) demographic characteristics of professionals (gender, age, marital status, years of work experience, education level, occupation, seniority) and workplace-related conditions [number of patients received per day, shift work, and anxiety experienced at work from 1 (absent) to 5 (extremely high) level]; (2) physical violence in the previous 12 months (defined as intentional injury to body); (3) situation of nonphysical violence in the previous 12months (including verbal abuse, threatening events, and sexual harassment); and (4).....

5.2 At page 11, line 57, I suggest to modify as follows:
All respondents who gave their informed consent filled....

Answer: According to your amendment, I have revised the questions mentioned above in the two parts (Page 11 line 14 to Page 12, Page 13 line 7, It has been marked yellow) , And the terms in the table have been modified (Page 24 to 32, It has been marked yellow). Details about this are as follows.

The final version of the questionnaire included four parts: (1) the demographic characteristics of the professionals, (gender, age, marital status, years of work experience, education level, occupation, seniority), and workplace-related conditions [number of patients received per day, shift work, number of co-workers for more than half of working hours, and anxiety experienced at work, (which was rated from 1, (absent), to 5, (extremely high)); (2) experience of physical violence in the previous 12 months, (defined as intentional injury to body); (3) experience of nonphysical violence over the previous 12 months, (including verbal abuse, threatening events, and sexual harassment);.....

..... confidentiality. All respondents who gave their informed consent completed the.....

(6) At page 11, lines 3-19, I suggest the following modifications:
“Depression Scale (SDS), which was able to reveal subjective feelings using 20 items with a severity score ranging from 1 (absent or minimum) to 4 (high or extremely high). The total score was recorded as the original score, then the scores of the 20 items were added, and finally the resulting score was multiplied by 1.25 to get the standard score. A final score of ≥ 53 points indicated the presence of depressive symptoms.”

Answer: According to your suggestion, I have modified this part. On the basis of your recommendation, we added the range of SDS scale and the use of this scale in this field. (Page 12 line 1, It has been marked yellow) Details about this are as follows.

(4) depressive symptoms, measured using a Chinese version of the Zung Self-Rating Depression Scale, (SDS), which has good reliability. SDS is usually used to measure the level of depressive symptoms in the general population over the previous week, and is widely used in research into WPV and depressive symptoms. Specifically, it facilitates the revealing of subjective feelings using 20 items which are rated in terms of severity using a four-point scale ranging from 1, (absent or minimum), to 4, (high or extremely high). The total score was recorded as the original score, the scores for the 20 items were added together, and finally the resulting score was multiplied by 1.25 to get the standard score. A standard score of 53~62 points represents mild depression, 63~72 points represents moderate depression, and >72 points signifies severe depression. To ensure that the results of this study can be compared with the results of related studies, we chose to use the same classification method as other studies: setting a final score of ≥ 53 points as indicating the presence of depressive symptoms.^{31, 32}

(7) In the Results section, I advise the following changes:

7.1 At page 13, line 7, “In Table 1, the demographic characteristics of the study participants are shown.”

7.2 At page 13, lines 14-19, “...and 78.2% were graduated or specialized. Of the respondents, 19.5% was senior within the department.”

7.3 At page 13, line 24, “...shown in Table 2. 57.2% of the 652 otorhinolaryngology physicians..”

7.4 At page 13, lines 26-29, “...with one or two colleagues for more than half of their working hours.”

7.5 At page 13, lines 46-49, “In Table 3, the variables related to depressive symptoms at the multiple logistic model, with the adjusted odds ratio (OR) and 95% confidence interval (CI), are shown. Our otorhinolaryngology physicians and nurses who...”

7,6 At page 14, lines 5-20, I suggest the following changes:

“Otorhinolaryngology physicians and nurses who worked alone for more than half of their working hours were more likely to suffer from depressive symptoms than those who worked frequently with other colleagues. Relative to the seniority of otorhinolaryngology physicians and nurses, formal staff and those with junior and intermediate titles showed protective factors for depressive symptoms.

Answer: According to your suggestion, we have made a careful revision of the results section.(Page 13 to 14, It has been marked yellow)

(8) It is not clear the meaning of “formal staff”

Answer: We have thought carefully about the question you raised, details about this are as follows. Our research study population is otolaryngology physicians and nurses who are from The Grade A tertiary hospitals. The informal staffs refers to The Grade A tertiary hospitals’ training physicians and nurses, this part of the staffs together with other staffs in the hospital to provide medical services for patients. According to the provisions of Chinese health policy, high level hospitals (e.g. The Grade A tertiary hospitals) have the training obligation to help staffs in lower level hospitals(e.g. County Hospital, Community Health Centre or Township Health Center), staffs in lower level hospitals should receive regular training at high level hospitals (Work together with staffs at a high level hospital to form a team). The training period is usually 3 months, 6 months or one year, this part of the medical staff s are not formal staffs.

(9) In the Discussion section, at page 15, lines 54-57 and page 16, lines 3-6, I suggest to cancel the following sentence “In China, the otorhinolaryngology department is a significant and indispensable group within the hospital, as it is related to patients’ ear, nose, throat, and even the head and neck.” because too generic and not specific. I suggest to modify the following sentence: “Unfortunately, in China, the frequency of violence within otorhinolaryngology department is relatively high due to various factors involving....”

Answer: According to your suggestion, I cancel the sentence “In China, the otorhinolaryngology department is a significant and indispensable group within the hospital, as it is related to patients’ ear, nose, throat, and even the head and neck.” And modify the following sentence:” Unfortunately, in China, the frequency of violence within otorhinolaryngology department is relatively high due to various factors involving treatment options and professional skill levels”(Page 15 line 16, It has been marked yellow) details about this are as follows.

..... China. Unfortunately, in China, the frequency of violence (which is a major predictor of depression) within otorhinolaryngology department is relatively high due to various factors involving treatment.....

(10) At page 18, line 19 the word “employee” as adjective for anxiety should be cancelled.

Answer: According to your suggestion, I cancel the word “employee”.(Page 17 line 9, It has been marked yellow)

(11) 11.1. At page 18, lines 46-57, I suggest the following changes: “In previous studies, the incidence of depressive symptoms was associated with physicians and nurses’

11.2. workload and shift characteristics; however, this study did not find any relevant association between depression and these two factors among otorhinolaryngology professionals [46-49], as well as another study of resident physicians in Turkey [50].” In this regard, I suggest to add the following citation (page 18, line 51, after “shift characteristics”), to report in the References: Ferri P, Guadi M, Marcheselli L, Balduzzi S, Magnani D, Di Lorenzo R.

The impact of shift work on the psychological and physical health of nurses in a general hospital: a comparison between rotating night shifts and day shifts. Risk Manag Healthc Policy. 2016 Sep 14;9:203-211.

Answer: According to your suggestion, I have revised the original sentence and added the references you recommended. (Page 18 line 19, It has been marked yellow) details about this are as follows.

.....In previous studies, the incidence of depressive symptoms was associated with physicians and nurses' workload and shift characteristics49-53...

(12) "Third, this study investigated only the association of workplace violence-related factors with depression without analyzing additional factors related to the occurrence of depressive symptoms." I suggest to cancel the following sentence because too speculative : "Nevertheless, the current results suggest that otorhinolaryngology physicians and nurses are more susceptible to depressive symptoms than other healthcare professionals and that certain risk factors associated physicians and nurses are more susceptible to depressive with workplace violence have a direct effect on this occurrence."

Answer: According to your suggestions and the content of this study, we re-combed the limitations of this part. Physicians and nurses who were absent from work because of depression were not assessed, the reporting rate of depressive symptoms may be lower than the true value are added. And modified the last sentence. The details are as follows.

This study has the following limitations. First, we only investigated otorhinolaryngology physicians and nurses working in Grade A tertiary hospitals in Heilongjiang Province, so the results may not be fully representative of the entire situation in northern China, and cannot be extended to other types of hospitals or departments. Future studies should conduct in-depth exploration of the different types of otorhinolaryngological services at hospitals of different grades. Second, physicians and nurses who were absent from work because of depression were not assessed; thus, the reported rate of depressive symptoms may be lower than the true value. Third, because the respondents were asked to review the events of the past 12 months, the prevalence of violence could have been under-reported as a result of reporting or recall bias or stigma and shame. Fourth, this study focuses on the more distinctive WPV-related factors for this analysis; however, there may be additional factors related to the occurrence of depressive symptoms that could be further explored. Nevertheless, the results of this study indicate that physical violence and mainly working alone are risk factors for depressive symptoms among otorhinolaryngology physicians and nurses, and this provides some basis and direction for future research.

(13) 13.1 In the Conclusion section, At page 19, line 39: I suggest to modify the sentence as following: "In conclusion, .."

13.2 In Table 2, page 32, line 9, the term "Title" should be changed to "Seniority"

In Table 2, page 32, line 19, the term "formal doctor" is not clear.

In Table 2, page 33, line 31, "A person" should be changed to "1".

In Table 2, page 33, lines 21-29, "Number of colleagues during work" could be changed to "Number of coworkers".

In Table 2, page 33, line 46, "Zero" should be changed to "Absent".

In Table 3, I suggest to modify the terms as above reported.

Answer: According to your suggestion, the conclusion(Page 19 line 18, It has been marked yellow)and the table of the article(Page 24 to 32, It has been marked yellow) have been revised.

I don't know if I understand your proposal correctly. If you have any other questions, please do not hesitate to contact me.

With best wishes,

Your sincerely,

Huiying Fang

Harbin Medical University

E:565453428@qq.com

Replies to Reviewer 2:

Dear Prof. Andréa Tenório Correia da Silva,

First of all, thank you for your earnest and constructive sincere comments on this article. According to your suggestion, I made a careful revision of the article, and gave a detailed answer to the questions you asked. I don't know if I understand your proposal correctly. If you have any other questions, please do not hesitate to contact me. Thank you! Here's the details. (The modified part of the manuscript is marked with yellow.)

(1) This cross-sectional study investigated depressive symptoms among otorhinolaryngology nurses and physicians in tertiary hospitals and their associations with work-related variables, including exposure to workplace violence. It is one of few studies of otorhinolaryngology nurses and physicians in tertiary hospitals. However, the study does not offer any unique insights into job related factors and their relationship with depressive symptoms. The authors should point out what the novelty of the study is, and emphasize how it fills in knowledge gaps. It is essential to explain why the approach is new and important.

Answer: Thank you very much for your questions and suggestions on this research. According to your questions and suggestions, We have carefully considered and revised the manuscript's abstract, introduction, conclusion and other parts to emphasize the creativity of this research. First, this research deals with an emerging issue of clinical practice. Depressive symptoms and workplace violence-related risk factors among otorhinolaryngology nurses and physicians. Previous studies focused on general departments, emergency departments and pediatrics, however, otolaryngology as an important department, almost no attention was paid to otolaryngology. Second, the result of our study is different from the general department, which indicates that the frequency of depressive symptoms among otorhinolaryngology physicians and nurses may be influenced by physical violence, number of coworkers for more than half of the working hours and other workplace violence-related factors. This reminds managers to strengthen the attention of Otolaryngology Department. Medical institutions should take effective measures to prevent the occurrence of physical violence, strengthen team cooperation ability, and increase peer support, to reduce the depressive symptoms caused by workplace violence and improve the quality of medical services. The detailed revises are as follows.

① We have made a detailed revision of the Objectives and conclusion of the abstract, emphasizing the innovation of this research: (Page 3 to 4, It has been marked yellow)

Objectives: Workplace violence is relatively frequent among medical professionals who work in otorhinolaryngology units. This phenomenon not only reduces the quality of provided medical care, but also increases the incidence of depressive symptoms among physicians and nurses, seriously affecting their job satisfaction and work efficiency with a negative attitude toward providing treatment. Few existing studies have assessed workplace-violence-related factors associated with depressive symptoms among otorhinolaryngology physicians and nurses.

Conclusions: This research addresses an emerging issue of clinical practice, and its results differ from those of previous studies; specifically, it indicates that the frequency of depressive symptoms among otorhinolaryngology physicians and nurses may be influenced by physical violence, the number of co-workers they have for more than half of their working hours, and other workplace-violence-related factors. To reduce the depressive symptoms caused by workplace violence and improve the quality of medical services, medical institutions should implement effective measures to prevent the occurrence of physical violence, strengthen team cooperation ability, and increase peer support.

②In the introduction, we emphasize the uniqueness of Otorhinolaryngology, and less research on otolaryngology, which highlights the innovation of this study.(Page 8 line 10,Page 9 line 8, It has been marked yellow)

Further, it should also be noted that much of the existing research primarily assesses the overall situations of hospitals, with few studies focusing on specialised departments. Of the latter studies, the focus has primarily been placed on the emergency department; however, as mentioned above, the department of otolaryngology witnesses a similar number of violent phenomena as emergency rooms, and more than that found in other departments; therefore, the occurrence of depressive symptoms among otolaryngology staff should be taken seriously.

few existing studies have assessed WPV-related factors associated with depressive symptoms among otorhinolaryngology physicians and nurses.

③In conclusion, we emphasized that the incidence of depressive symptoms was higher in the otolaryngology physicians and nurses, and that the influencing factors were different from those in the general department, and that the targeted measures could be taken to change the phenomenon. To highlight our innovative and practical significance.(Page 19 line 18, It has been marked yellow)

In conclusion, the results show that otorhinolaryngology physicians and nurses in Grade A tertiary hospitals in Heilongjiang Province, China, have a high risk of experiencing depressive symptoms. Of the 652 respondents, 57.2% reported varying degrees of depressive symptoms. Our findings also show that experiencing physical violence, number of co-workers for more than half of working hours, and other WPV-related factors are associated with symptoms of depression.

(2) The objectives of the study should be more accurately described. The way it is displayed throughout the manuscript is inconsistent and may lead the reader to uncertainty about whether the objective was: (1) to investigate the association between exposure to workplace violence and depressive symptoms; (2) to examine variables associated with workplace violence; (3) or to assess factors associated with depressive symptoms, including workplace violence. For instance, in the abstract “to investigate the relationship between workplace violence-related risk factors and depressive symptoms”, and in the introduction “we found few previous studies on Chinese otorhinolaryngology physicians and nurses suffering from depression-related factors, including violence. This study seeks to assess the level of depression... and to identify associations among depressive symptoms and workplace violence-related risk factors”.

Answer: Thank you very much for your question. We made a little mistake in language. We have revised the whole text. Thank you for your carefulness. Specific modifications are as follows.

(Page 9 line 11, It has been marked yellow), At the end of the introduction, we clearly put forward the purpose of this study:” Consequently, the purpose of this study is: (1) to investigate the prevalence of depressive symptoms among otorhinolaryngology physicians and nurses; and (2) to assess the risk factors that influence depressive symptoms, including WPV-related risk factors and demographic variables, and to present some specific suggestions.”

(3) With respect to the methods section, the questionnaire used to measure depression can make comparison with other studies limited. Moreover, it is necessary to explain the use of the Zung Self-Rating Depression Scale to assess both depression and anxiety, and classifying participants in levels of anxiety (zero, low, moderate, high, and extremely high). In addition, since this is a cross-sectional study, it is not possible to examine prediction. My suggestion to the authors would be to review this section according to the items in the STROBE Statement. This will probably make it easier for the audience to understand what the authors have done step-by-step.

Answer: I don't know if I have understood your suggestion correctly. If I don't understand it correctly, please don't be tired of contact with me.

According to my understanding of your suggestion, I made a detailed revision of the SDS scales' introduction. Firstly, the application scope of SDS scale was introduced. Second, it is pointed out that this scale is often used in other studies of this field, the results of the study can be compared with the results of others. Third, the scoring rules and results classification of SDS scale are introduced in detail. Finally, the classification methods of the results are given. Specific modifications are as follows (Page 12 line 1, It has been marked yellow):

(4) depressive symptoms, measured using a Chinese version of the Zung Self-Rating Depression Scale, (SDS), which has good reliability. SDS is usually used to measure the level of depressive symptoms in the general population over the previous week, and is widely used in research into WPV and depressive symptoms.^{8,30} Specifically, it facilitates the revealing of subjective feelings using 20 items which are rated in terms of severity using a four-point scale ranging from 1, (absent or minimum), to 4, (high or extremely high). The total score was recorded as the original score, the scores for the 20 items were added together, and finally the resulting score was multiplied by 1.25 to get the standard score. A standard score of 53~62 points represents mild depression, 63~72 points represents moderate depression, and >72 points signifies severe depression. To ensure that the results of this study can be compared with the results of related studies, we chose to use the same classification method as other studies: setting a final score of ≥ 53 points as indicating the presence of depressive symptoms.^{31, 32}

(4) The limitations should include: (1) physicians and nurses who were absent from work because of depression were not assessed, and (2) the prevalence of violence could have been under-reported due to reporting or recall bias, or stigma and shame.

Answer: Thank you very much for your comments. We think your comments are very useful. Therefore, in combination with your opinion, we have made a detailed revision of the limitations. Specific modifications are as follows. (Page 19 line 7, It has been marked yellow)

Second, physicians and nurses who were absent from work because of depression were not assessed; thus, the reported rate of depressive symptoms may be lower than the true value. Third, because the respondents were asked to review the events of the past 12 months, the prevalence of violence could have been under-reported as a result of reporting or recall bias or stigma and shame.

(5) Finally, the conclusion should be more focused on the actual results. The main finding was the association of exposure to physical violence and depressive symptoms. The authors suggested that intervention should include improving team relationships. Nevertheless, the authors have not addressed workplace conflicts (such as bullying, harassment, physical violence between staff etc.), or social support from colleagues and supervisors (e.g. Demand-Control-Support model).

Answer: According to your suggestion, we have revised the discussion and conclusion. The newly added part focusing on the importance of teamwork, and the measures to prevent physical violence. Specific modifications are as follows.

① In the discussion section, based on the demand-control-support model, we point out the role of peer support in alleviating depressive symptoms, and propose that peer support can be enhanced through reinforcement of team cooperation to alleviate depressive symptoms. Detailed modifications are as follows. (Page 16 line 21, It has been marked yellow)

Support at work is a social-network resource associated with the work environment, and employees at all levels can benefit from the support of their colleagues. In the demand-control-support model, it has been highlighted that social support is similar to job control, and may have a buffering or reinforcing effect on individuals' psychological responses; for example, work friendship between colleagues has been reported to be an important social support.⁴⁴ Therefore, the working status of medical staff, (e.g. time spent working alone), should be considered in order to strengthen peer support, (as well as team cooperation), and thereby effectively prevent and improve depressive symptoms.

② In the discussion section, physical violence is one of the most important factors affecting depressive symptoms, We suggest that medical institutions and managers should take corresponding measures to prevent physical violence. Detailed modifications are as follows. (Page 18 line 15, It has been marked yellow)

Concurrently, medical institutions should adopt a 'zero tolerance' approach to violent incidents and, to reduce the harm caused by physical violence, managers should strengthen the provision of timely psychological comfort and psychological counselling for employees who have experienced physical violence.

③ Finally, we have made a systematic revision of the conclusion. On the one hand, we emphasize the need to strengthen team capacity. On the other hand, we point out that zero tolerance policy should be adopted for physical violence. Finally, it is pointed out that managers should pay attention to horizontal violence. Detailed modifications are as follows. (Page 19 line 18, It has been marked yellow)

In conclusion, the results show that otorhinolaryngology physicians and nurses in Grade A tertiary hospitals in Heilongjiang Province, China, have a high risk of experiencing depressive symptoms. Of the 652 respondents, 57.2% reported varying degrees of depressive symptoms. Our findings also show that experiencing physical violence, number of co-workers for more than half of working hours, and other WPV-related factors are associated with symptoms of depression. Therefore, on one hand, it is clear that medical institutions should pay more attention to otorhinolaryngology. Essential measures include improving communication between medical staff and patients and the adoption of a 'zero tolerance' attitude towards hospital violence. On the other hand, attention should be paid to the construction of team relationships, improving teams' cooperation, and giving full backing to peer support to alleviate depressive symptoms. Finally, more attention should also be paid to lateral violence, (bullying, harassment, physical violence between staff, etc.) to address depressive symptoms caused by such violence. However, as we did not study the impact of lateral violence on depressive symptoms, this represents an avenue of research for future studies.

I don't know if I understand your proposal correctly. If you have any other questions, please do not hesitate to contact me.

With best wishes,

Your sincerely,

Huiying Fang

Harbin Medical University

E:565453428@qq.com

VERSION 2 – REVIEW

REVIEWER	Ferri Paola University of Modena and Reggio Emilia
REVIEW RETURNED	11-Nov-2017

GENERAL COMMENTS	The manuscript “Depressive symptoms and workplace violence-related risk factors among otorhinolaryngology nurses and physicians in northern China: A cross-sectional study” has been carefully revised by the authors according to my suggestions. I think that it needs only some minor revisions before publishing indicated in the attached file. The manuscript “Depressive symptoms and workplace violence-related risk factors among otorhinolaryngology nurses and physicians in northern China: A cross-sectional study” has been carefully revised by the authors according to my suggestions. I think that it needs only some minor revisions before publishing, here indicated: 1. in the paragraph Results of the Abstract, I suggest to change “57.2% were found” to “57.2% was found”;2. in the Methods to put the first letter of subheadings in capital letter (e.g. 2.1 Study population, 2.2 Data collection, 2.3 The Questionnaire preparation, 2.4 Data analysis, 2.5 Ethics statement);3. in the Results, page 14 lines 3-4, I suggest to change “78.2% were graduated” to “78.2% was graduated”;4. the reference number 5 has to be corrected: “Paola, F., et al.,” to “Ferri, P., et al.”;5. the reference number 29 has to be corrected and the following link should be added as follows: ILO/ICN/WHO/PSI, Workplace Violence in the Health Sector Country Case Study-Questionnaire. 2003: p. 1-14. http://www.who.int/violence_injury_prevention/violence/interpersonall/en/WVquestionnaire.pdf;6. the reference number 53 has to be corrected: “Paola, F., et al.,” to “Ferri, P., et al.”; “p. 203.” to “p. 203-211.”.
--

VERSION 2 – AUTHOR RESPONSE

Replies to Reviewer 1:

Dear Prof. Ferri Paola,

First of all, thank you for your earnest and constructive sincere comments on this article. According to your suggestion, I made a careful revision of the article. If you have any other questions, please do

not hesitate to contact me. Thank you! Here's the details.(The modified part of the manuscript is marked with yellow.)

1. in the paragraph Results of the Abstract, I suggest to change "57.2% were found" to "57.2% was found".

Answer: According to your revised opinion, we revised the manuscript. (In the paragraph Results of the Abstract, It has been marked yellow) Details about this are as follows.

.....57.2% was found to have depressive symptoms,.....

2. in the Methods to put the first letter of subheadings in capital letter (e.g. 2.1 Study population,2.2Data collection, 2.3 The Questionnaire preparation, 2.4 Data analysis, 2.5 Ethics statement);

3. in the Results, page 14 lines 3-4, I suggest to change "78.2% were graduated" to "78.2% was graduated".

Answer: According to your suggestion, we have made a careful revision of the Methods and Results section. (In the Methods, and first paragraph of the Results. It has been marked yellow) Details about this are as follows.

In the Methods section. 2.1. Study population, 2.2 Data collection, 2.3. Questionnaire preparation,2.4. Data analysis, 2.5. Ethics statement

In the Results section.had at least 10 years of experience in otolaryngology and 78.2% was graduated or specialised.

4. the reference number 5 has to be corrected: "Paola, F., et al.," to "Ferri, P., et al.,";

5. the reference number 29 has to be corrected and the following link should be added as follows:ILO/ICN/WHO/PSI, Workplace Violence in the Health Sector Country Case Study-Questionnaire.2003: p. 1-

14.http://www.who.int/violence_injury_prevention/violence/interpersonal/en/WVquestionnaire.pdf;

6. the reference number 53 has to be corrected: "Paola, F., et al.," to "Ferri, P., et al.,"; "p. 203." to "p.203-211.".

Answer: Thank you for your suggestion. According to your suggestion, we have revised the references number 5, 29 and 53. (It has been marked yellow) Details about this are as follows.

[5]. Ferri, P., et al., Workplace violence in different settings and among various health professionals in an Italian general hospital: a cross-sectional study. *Psychology Research & Behavior Management*, 2016. 9: p. 263-275.

[29]. ILO/ICN/WHO/PSI, Workplace Violence in the Health Sector Country Case Study-Questionnaire. 2003: p. 1-

14.http://www.who.int/violence_injury_prevention/violence/interpersonal/en/WVquestionnaire.pdf.

[53]. Ferri, P., et al., The impact of shift work on the psychological and physical health of nurses in a general hospital: a comparison between rotating night shifts and day shifts. *Risk Management & Healthcare Policy*, 2016. 9: p. 203-211.

Thank you again for your patient suggestions for this article. If you have any other questions, please do not hesitate to contact me.

With best wishes,

Your sincerely,

Huiying Fang

Harbin Medical University

E:565453428@qq.com